# Effect of Work–Family Conflict, Psychological Job Demand, and Job Control on the Health Status of Nurses

**DOI:** 10.3390/ijerph18073540

**Published:** 2021-03-29

**Authors:** Li-Chung Pien, Wan-Ju Cheng, Kuei-Ru Chou, Li-Chiu Lin

**Affiliations:** 1Post-Baccalaureate Program in Nursing, College of Nursing, Taipei Medical University, 250 Wu-Hsing Street, Taipei 11031, Taiwan; andy5240@tmu.edu.tw; 2Psychiatric Research Center, Wan Fang Hospital, Taipei Medical University, No. 111, Sec. 3, Xinglong Rd., Wenshan District 116, Taipei 11608, Taiwan; 3Department of Psychiatry, China Medical University Hospital, No. 2, Yude Rd., North District 404332, Taichung 40447, Taiwan; s871065@gmail.com; 4Department of Public Health, China Medical University, No. 100, Sec. 1, Jingmao Rd., Beitun District 406040, Taichung 40402, Taiwan; 5School of Nursing, College of Nursing, Taipei Medical University, 250 Wu-Hsing Street, Taipei 11031, Taiwan; kueiru@tmu.edu.tw; 6Center for Nursing and Healthcare Research in Clinical Practice Application, Wan Fang Hospital, Taipei Medical University, No. 111, Sec. 3, Xinglong Rd., Wenshan District 116, Taipei 11608, Taiwan; 7Department of Nursing, Taipei Medical University-Shuang Ho Hospital, No. 291, Zhongzheng Rd., Zhonghe District, New Taipei City 23561, Taiwan; 8Psychiatric Research Center, Taipei Medical University Hospital, No. 252, Wuxing Street, Xinyi District, Taipei 110301, Taiwan; 9Nursing Department, Hung Kuang University, 1018 Taiwan Boulevard, Sec. 6, Shalu District, Taichung 433304, Taiwan

**Keywords:** work–family conflict, self-rated health, depression, leaving intention

## Abstract

Work–family conflicts (WFCs) are common in the healthcare sector and pose significant health risks to healthcare workers. This study examined the effect of WFCs on the health status and nurses’ leaving intentions in Taiwan. A self-administered questionnaire was used to survey 200 female nurses’ experiences of WFC from a regional hospital. Data on psychosocial work conditions, including work shifts, job control, psychological job demands, and workplace justice, were collected. Health conditions were measured using the Beck Depression Inventory-II and self-rated health. Leaving intentions were measured using a self-developed questionnaire. The participants’ average work experience was 6.79 (Standard Deviation (SD) = 5.26) years, their highest educational level was university, and work shifts were mostly night and rotating shifts. Approximately 75.5% of nurses perceived high levels of WFCs. Leaving intentions were correlated with WFCs (*r* = 0.350, *p* < 0.01) and psychological work demands (*r* = 0.377, *p* < 0.01). After adjusting for age, educational level, and work characteristics, high levels of WFCs were associated with poor self-rated health, and depression, but not associated with high leaving intentions. Nurses’ experiences of high levels of WFCs greatly affected their health status.

## 1. Introduction

Statistics of the Taiwan Union of Nurses Association (TUNA), Taiwan and Fujian Province of China, in September 2020 showed that Taiwan had 180,526 licensed and 170,240 nursing workers, of which 96.5% were women [1]. Currently, the nursing work force in Taiwan consists mostly of women. The nursing manpower shortage in Taiwan causes a relatively uneven nurse-to-patient ratio, high work pressure, and high work load, which result in increased overtime and turnover rates. Female workers have to fulfil family household responsibilities in addition to bearing workplace pressure, resulting in conflicts between family and work [1,2]. Especially, work–family conflicts (WFCs) are a common problem faced by nursing staff [3,4].

However, WFCs are not only an issue in Taiwan but also around the world; different types of literature have indicated [3,4,5,6,7,8] that working conditions are much more about WFCs. As employed women, nurses have to not only cope with workplace pressure but also take care of housework and children after completing work. Furthermore, research has shown that role conflicts between work and housework are the biggest sources of stress for women [9,10]. Therefore, work–family conflicts (WFCs) and occupational fatigue of female nursing staff are essential indicators and tools for their assessment. If medical institution managers understand the WFCs of married female nursing staff and improve their work conditions and environment, they can reduce the pressure, promote nurses’ physical and mental health, and improve the quality and effectiveness of care services.

## 2. Literature Review

Different types of literature show that WFC is an essential factor affecting workers’ physical and mental health [5,7,11]. Kahn et al. (1964) defined WFC as ‘’the conflict and pressure between roles caused by the incompatibility between work and family roles’’ [12]. Furthermore, a systematic review indicated out by Greenhaus and Buetell [13] in 1985 maintained that WFC undoubtedly reflects people’s belief that work and family life are interdependent. Hu et al. [6] also indicated that different norms and responsibilities are involved in work and family, and that the work role may make fulfilling family roles difficult, which leads to WFCs.

Moreover, some studies also stated that conflict and pressure result from competition between multiple roles [6,13]. For instance, in 2019, Galletta et al. [7] examined the moderating role of collective affective commitment as a protective resource in the relationship between WFC and emotional exhaustion in 2019. The study [7] found that emotional commitment can be regarded as a protective resource for nurses. Nurses with low family and work conflict have a higher investment in work. The researchers [7] recommend using interventions to reduce WFC, promote nurse’s commitment, and reduce emotional exhaustion. Conflict and pressure lead to decreased time and energy, making it difficult for workers to perform their roles, resulting in excessive workload and fatigue [5,6]. 

After investigating 351 full-time participants, the Zhou et al. [14] study suggested that WFC affected the level of self-reported mental health. Those researchers [14] indicated that conflict between workplace and family is not only a negative impact but also a predictor of mental health. Evidence also shows that WFCs are markedly related to worker health and work performance [5]. High levels of WFCs can cause physical and mental stress and emotional exhaustion [14,15,16]. In terms of physical health, poor self-rated health (SRH) [8,11]; depression [17,18,19,20]; problem drinking behavior [15,18]; and physical symptoms such as fatigue, lack of appetite, and nervous tension are positively correlated [15,21,22]. 

Some investigators have found that a favorable workplace enables workers to balance their work and family roles, thereby reducing WFCs [8,14,23,24,25]. However, studies also indicated that workplace characteristics such as low job control, high psychological work demands, night/rotating shifts, and low workplace justice are risk factors for the poor physical and mental health of workers [26,27,28]. Especially, with increased working hours, high pressure, and shift work, the nursing profession is associated with a high risk of WFCs [11,23,29]. In particular, compared with male nurses, female nurses face a high level of WFC [19]. In terms of work, WFCs can lead to poor quality of care provided by nursing staff [30] and increased turnover rates [30,31,32,33]. In 2013, Pien et al. [11] surveyed 17,892 participants to investigate the impact of WFC on mental health. The researchers found that heavy workload, low workplace justice, and family care burden were associated with WFC and subsequently with poor mental health [11].

In previous studies on WFCs, simultaneous consideration of the effects of work characteristics on the health of nursing staff is lacking. It is unclear whether the health effects of WFCs still remain significant after controlling for psychosocial work conditions. This research aimed to determine the effects of WFCs and major psychosocial work characteristics relevant to nurses on their health. We hypothesized that WFCs was associated with worse health among nurses after adjustment for psychosocial work conditions.

## 3. Methods

### 3.1. Study Population and Procedures

This cross-sectional study was conducted using standardized self-administered questionnaires. Convenience sampling of female nurses was performed. A total of 220 questionnaires were distributed, and 200 copies were retrieved and used for the final analysis the response rate was 90.9%. The inclusion criteria were female nurses with a full-time job, age ≥20 years, and employed by a hospital at the time of the survey. A trained research assistant of our research team administered research questionnaires to nurses.

### 3.2. Measurements

#### 3.2.1. WFC Assessment

The Chinese version of the five-item WFC scale, originally established by Netemeyer et al. [34], was used to investigate participants’ WFC status. The five items are as follows: (1) my work is demanding, which interferes with my home and family life; (2) my job hours make it difficult to fulfil my family responsibilities; (3) I am unable to do things I wish to at home because of my demanding job; (4) my job stress makes it difficult to fulfil my family duties; and (5) work-related duties lead to changes in my plans regarding family activities. Cronbach’s α of the Chinese version of the WFC scale is 0.95 [11]. Each item is scored from 1 (strongly disagree) to 5 (strongly agree), with the total score ranging from 5 to 25. An average score is calculated by summing the scores of the five items, with a high score indicating a high level of WFC. The average scores are further dichotomized into a high level of WFC (score of >3) and low level of WFC (score of ≤3).

#### 3.2.2. Health Status Assessment

SRH of nurses was assessed based on a single question, ‘’In general, how is your health?’’, and the answer was recorded on a 5-point scale, ranging from 1 (very good) to 5 (very poor). In this study, responses were dichotomized into poor SRH (poor or very poor) and good SRH (moderate, good, or very good).

For assessing nurses’ mental health, the 21-item Beck Depression Inventory-II was used; its Chinese version questionnaire was validated, and Cronbach’s α of the scale was 0.93 [35,36]. This questionnaire contained 21 questions, with each question having four possible answers that were sorted according to depressive symptom severity; the scores range from 0 to 63. A high score indicated severe depression symptoms, and a summed score of >14 indicated depression problems [35].

#### 3.2.3. Assessment of Leaving Intentions

The nurses’ turnover intention scale was established based on previous studies [37,38]. Questions included (1) ‘’How often have you thought about leaving your current position?’’, (2) ‘’How often have you thought about finding other nursing jobs?’’, (3) ‘’How often have you thought about finding other nursing-related jobs?’’, and (4) ‘’How often have you thought about giving up the nursing profession completely and starting a different type of job?’’ Answers were recorded on a 5-point scale ranging from 1 (never) to 5 (always). Analyses showed high internal consistency of these four items, with Cronbach’s α of 0.90 and high construct reliability (model fit indexes were as follows: χ^2^/df = 1.450, GFI = 0.995, AGFI = 0.973, IFI = 0.999, TLI = 0.997, and CFI = 0.999). The summed score was standardized to the 0–100 range, and a standardized score of >50 indicated high leaving intentions.

#### 3.2.4. Assessment of Other Psychosocial Work Characteristics

The psychosocial work characteristics scale consisted of 22 items: job control (nine items), psychological job demands (five items), workplace justice (seven items), and shift work (one item). Psychological job demands and job control were assessed using the Chinese version of the Job Content Questionnaire based on Karasek’s job strain model [28,39,40,41,42,43].

The job control scale consisted of two subscales: skill discretion subscale (six items: learning new things, a high level of skills, nonrepetitive work, creative work, various tasks, and developing one’s abilities) and decision authority subscale (three items: allowed to make own decisions, freedom to make decisions, and opinions influential). Job demands were assessed using five items related to psychological concerns (work fast, excessive work, not enough time, concentrate on job for a long time, and very busy). Workplace justice was assessed using seven items (trust, information reliable, work arranged fairly, rewards arranged fairly, performance evaluated fairly, information during the decision-making process, and respect). During data analysis, job control, psychological work demands, and workplace justice were divided into low, medium, and high according to the scores, and these were included in the regression model. Both scales showed favorable psychometric properties in previous studies [39,40]. Responses for all the aforementioned items were recorded on a 4-point Likert scale, ranging from 1 (strongly disagree) to 4 (strongly agree).

Participants were asked whether they had fixed shifts or night/rotating shifts during the week before the survey, and those with a fixed day shift were classified as day shift workers, whereas the others with night/rotating shifts were classified as night/rotating shift workers.

#### 3.2.5. Ethical Considerations

This study was reviewed and approved by the Institutional Review Board (IRB) of Tungs’ Taichung Metro Harbour Hospital (IRB no.: TTMHH101004).

#### 3.2.6. Data Analysis

Descriptive statistics were performed on the distributions of sociodemographic characteristics, WFCs, health status, leaving intentions, and working conditions. Multivariate logistic regression was performed using SPSS 24.0 (IBM, Armonk, NY, USA) to examine the relationship of WFCs with nurses’ health outcomes and leaving intentions after adjusting for working conditions.

## 4. Results

Table 1 summarizes the demographic characteristics, working conditions, leaving intentions, and health status of the study participants and the distribution of WFCs score (mean = 3.66, standard deviation = 0.77). Additionally, Pearson correlations showed that WFCs were significantly and positively associated with psychological work demands (*r* = 0.377, *p* < 0.01) and leaving intentions (*r* = 0.350, *p* < 0.01) (Table 2).

Table 3 shows the results of the multivariate logistic regression analysis of three health outcomes and leaving intentions. The findings of the regression models indicated that a high level of WFC was a significant predictor of poor self-rated health (odds ratio (OR) = 2.819) and depression (OR = 2.809), but not of high leaving intentions, even after adjusting for age, educational level, and adverse psychosocial work factors.

## 5. Discussion

The score of female nursing staff’s WFC score was higher in this study than in a previous national survey in Taiwan [11]. Moreover, most female nursing staff working in hospitals have high levels of WFCs, which is consistent with the results of previous studies [11,19,29]. In Taiwan, most nurses are of childbearing age. Prolonged working hours, irregular shifts, and high workloads create a high risk of poor SRH. For nurses, the high workload combined with inadequate family-friendly workplace policies and practices may be the root cause of WFCs, poor health of workers, depression, and their intention to leave.

The findings showed that high job requirements, and willingness to leave, were related to WFCs, which is consistent with the results of previous studies [10,11,18]. However, a closer examination of various research results showed that female nurses had high levels of WFCs. Because of their traditional roles in the family, female nurses may be more susceptible to physically harsh working conditions and clinical nursing work characteristics. Further investigation is needed to determine sex-dependant social norms and family-friendly work environments in the workplace and at home and how WFCs are differently experienced [15,19,25].

Inferential statistics after controlling for age, educational level, work shifts, and workplace justice showed that nurses with a WFC score of >3 had considerably high risks of poor SRH and depression. These results are consistent with previous research results, showing that job demands can synergistically increase stress-related health risks [13,15,18,39,44], but not leaving intentions. This study showed that WFCs did not significantly negatively affect turnover intentions, which is inconsistent with previous reports [30,31,33], and this have been caused by the “healthy worker effect”. The ‘’healthy worker effect’’ causes sample selection bias, as those severely affected by WFCs may have already left the workplace and thus not been included in the study [45,46]. Therefore, authors should be cautious when inferring the negative effects of WFCs.

This study has some limitations. First, the cross-sectional survey design precluded direct causal inferences. Causal inferences were restricted to the influence of WFCs on the physical and mental health and turnover intentions of nursing staff. In addition, reverse causality may occur. For example, nursing staff with poor physical and mental health may have poor work performance and be more likely to experience WFCs, or when workers’ physical and mental health is poor, they may perceive increased WFCs, thus influencing research inferences.

Second, participants in the study were current employees; thus, the results of this study could not be generalized to previous employees who may have faced a high risk of WFCs and have already left the workplace. In addition, the self-selection effect might exist that workers’ who had lower WFCs selected this job or remained working as nurses. Our results may have been biased by this healthy worker effect, and therefore the association between high WFC and leaving intentions may have been underestimated. Further research is needed to study WFCs in groups that have already resigned to determine their reasons for leaving.

Third, health outcome measurement was based on self-reports, which may be biased because of participants’ subjective perceptions or feelings. In future, objective measures (e.g., cortisol or other biomarkers) must be used for this type of research.

Fourth, although confounding factors such as demographic characteristics, job changes, and job-related characteristics were controlled for in our statistical model, other potential factors such as an individual’s socioeconomic status and family and workplace characteristics were not considered, including support level and other sources of life stress [11,14,29,47]. Organizational factors such as family-friendly policies and practices may confound or change the connection between workers’ experience of WFCs and their negative effects. Therefore, these organizational factors should also be included in future studies.

Despite these limitations, the study results are still relevant. The authors found that conflicts between work and family are common among nurses and create additional health risks. To improve the health of nurses, the government should strengthen family-friendly policies in the workplace. Specific measures include limiting excessive working hours, regulating the working hours of night shift workers, reducing work-load, and imposing higher penalties for employers who violate legal working conditions. Finally, both institutions and occupational health professionals must identify high risk factors and implement measures to reduce conflicts between work and family life. At an individual level, we advocate that awareness of WFCs’ impact on health should be increased [48], especially among married women. At a household level, support from family members, and on-demand child care services for employed woman, are crucial for reducing WFCs.

## Figures and Tables

**Table 1 ijerph-18-03540-t001:** Demographic characteristics of study participants (*N* = 200).

Variable	Mean/*n*	SD/%
Work experience (years)	6.79	5.26
Age (years)	28.50	5.29
20~30	143	71.5%
≥31	57	28.5%
Educational level		
Nursing high school and junior college	93	46.5%
University and above	107	53.5%
Work shift		
Fixed day shift	25	12.5%
Night and rotating shift	175	87.5%
Work-family conflict	3.66	0.77
Low (≤3)	49	24.5%
High (>3)	151	75.5%
Job control		
High	53	26.5%
Medium	81	40.5%
Low	66	33.0%
Psychological work demands		
High	79	39.5%
Medium	54	27.0%
Low	67	33.5%
Workplace justice		
High	59	29.5%
Medium	60	30.0%
Low	81	40.5%
Depression		
Normal	105	52.5%
Mild depression	37	18.5%
Moderate depression	32	16.0%
Severe depression	26	13.0%
Self-reported health		
Good	138	69.0%
Poor	62	31.0%
Leaving intentions		
Low	91	45.5%
High	109	54.5%

SD, standard deviation.

**Table 2 ijerph-18-03540-t002:** Correlation of work–family conflicts (WFCs) with job control, psychological work demands, and leaving intentions (*N* = 200).

Variable	1	2	3	4
1. WFCs	1			
2. Job control	−0.135	1		
3. Psychological work demands	0.377 **	0.076	1	
4. Leaving intentions	0.350 **	−0.224 **	0.318 **	1

** *p* < 0.01.

**Table 3 ijerph-18-03540-t003:** Odds ratio (OR) for poor self-rated health (SRH), depression, and high leaving intentions of nurses (*N* = 200).

Variable	Poor SRH		Depression		High Leaving Intentions	
	OR	95% CI	OR	95% CI	OR	95% CI
High WFC (ref. low)	2.819 *	(1.140, 6.972)	2.809 **	(1.329, 5.939)	1.634	(0.793, 3.370)
Job control						
High	1		1		1	
Medium	1.290	(0.578, 2.877)	0.833	(0.392, 1.773)	1.626	(0.748, 3.535)
Low	0.944	(0.412, 2.163)	1.076	(0.496, 2.336)	2.756 *	(1.222, 6.215)
Psychological work demands						
Low	1		1		1	
Medium	1.328	(0.536, 3.292)	0.388 *	(0.171, 0.882)	1.624	(0.733, 3.596)
High	2.630 *	(1.120, 6.175)	0.849	(0.391, 1.845)	3.777 **	(1.677, 8.504)

* *p* < 0.05; ** *p* < 0.01; Age, educational level, shift work, and workplace justice were controlled for in the regression model; WFC, work–family conflict; CI, confidence interval.

## Data Availability

The data presented in this study are available on request from the corresponding author. The data are not publicly available due to the information is related to personal privacy information, so it is not convenient to provide.

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
