# Peer review of "Effect of Work–Family Conflict, Psychological Job Demand, and Job Control on the Health Status of Nurses"

_ijerph, 2021, doi:10.3390/ijerph18073540_

Round 1
Reviewer 1 Report
Article: Effect of Work–family Conflict, Psychological Job Demand, and 2 Job Control on the Health Status of Nurses
This article examines examined the effect of Work–family conflicts (WFCs) on the health status 25 and nurses’ leaving intentions. They use a survey with information from 200 female nurses. This information is related to their psychosocial work conditions, work shifts, job control, psychological job demands, and workplace justice. They found that high levels of WFCs were associated with poor self-rated health, depression, but not associated with high leaving intentions. The paper has a correct literature review, and the methodology is adequate and consistent with the goals of the study. In any case, we have to bear in mind that this study is mainly descriptive, with a correlation analysis (Table 2) and a multivariate logistic regression. Nevertheless, the results are interesting and provide new insights on the main key variables. Also, the authors mentioned some limitations of the study: reverse causality, self-report measures, and other factors not considered within the analysis. Finally, I miss more public policy recommendations and suggestion at an individual and household level in the conclusion section.
Reviewer 2 Report
ijerph-1132090
Effect of Work–family Conflict, Psychological Job Demand, and Job Control on the Health Status of Nurses
- The abstract content appears to match the title. I encourage stating country of the work.
- In the introduction, please include something about culture of being able to leave/not leave a job. Many have written about work issues. Why was it important to examine your variables? Have others not examined them?
- The literature review lacks information about culture. Was culture part of any of the studies?
- Instruments are nicely described.
- In terms of sample, how many nurses were asked versus the 200 who participated? What was the response rate?
- IRB review is stated.
- Table 2 lacks column 4. Maybe it is cut off with shift of page???? The tables with the centering approach for words is difficult to read. I recommend the variable names be left justified.
- The discussion has new statistics. All statistics first must be in the results. Authors did relate variables to the literature. Why is your study unique? What did you add to the literature? Limitations are shared.
- Most references are older than 3-5 years but appear to match the topic.
Reviewer 3 Report
While this is an important topic with universal relevance, I found the paper lacking. Most significant is that there were no real hypotheses that emanated from the literature review. What was missing in previous research to make the undertaking of this study important? What would the results of this study contribute to our understanding. It appeared that only future research would address reasons for why the nurses in this study might not turnover, despite the job being detrimental to their mental health.
I would also be interested in knowing perhaps why these individuals entered the nursing profession to begin with. It is possible that they self select into the profession and that might be a powerful factor in their remaining on-the-job. Are there any realistic job previews during onboarding?
In any event, I feel that addressing the above fundamental questions for the study at hand would make the manuscript worthy of publication.
Round 2
Reviewer 1 Report
The authors have included all my suggestions and recommendations in this revised version.
Author Response
Thank you for your positive recognition.
Reviewer 3 Report
Authors addressed many of the concerns I had. While I saw additional sources added to the paper, I feel that the literature review should have spelled out more of what happened in the studies rather than just categorize them thematically and add a corresponding source number to the brackets.
